# Exogenous corticosterone and melanin-based coloration explain variation in juvenile dispersal behaviour in the barn owl (*Tyto alba*)

**Bettina Almasi** [1⦿]*, **Carolina Massa**[1,2⦿], **Lukas Jenni**[1], **Alexandre Roulin**[2]

**1** Swiss Ornithological Institute, Sempach, Switzerland, **2** Department of Ecology and Evolution, University of Lausanne, Lausanne, Switzerland

⦿ These authors contributed equally to this work.
* bettina.almasi@vogelwarte.ch

## Abstract

Natal dispersal affects many processes such as population dynamics. So far, most studies have examined the intrinsic and extrinsic factors that determine the distance between the place of birth and of first breeding. In contrast, few researchers followed the first steps of dispersal soon after fledging. To study this gap, we radio-tracked 95 barn owl nestlings (*Tyto alba*) to locate their diurnal roost sites from the fledging stage until December. This was used to test whether the age of nest departure, post-fledging movements and dispersal distance were related to melanin-based coloration, which is correlated to fitness-related traits, as well as to corticosterone, a hormone that mediates a number of life history trade-offs and the physiological and behavioural responses to stressful situations. We found that the artificial administration of corticosterone delayed the age when juveniles left their parental home-range in females but not in males. During the first few months after fledging, longer dispersal distances were reached by females compared to males, by individuals marked with larger black feather spots compared to individuals with smaller spots, by larger individuals and by those experimentally treated with corticosterone. We conclude that the onset and magnitude of dispersal is sensitive to the stress hormone corticosterone, melanin-based coloration and body size.

## Introduction

Natal dispersal, the movement between the places of birth and of first reproduction, has crucial effects on population biology, ecology and evolution [e.g. 1, 2]. The ultimate drivers of natal dispersal are the avoidance of inbreeding and the reduction of competition for nest sites and food [3, 4]. At a proximate level, the patterns of dispersal are diverse and species-specific [1, 5–8]. An individual's decision to disperse depends on extrinsic factors, such as environmental condition [9–11], predation risk [12, 13], social interactions [12, 14], sex ratio [reviewed in 15], and on intrinsic phenotypic traits such as sex [8, 16, 17], behavioural phenotypes [reviewed in 18] and colour morph [13, 19–21]. This last trait can be related to dispersal

**Funding:** The Swiss National Science Foundation supported financially the study (no 3100A0-104134 to LJ and no PPOOAO-102913 to AR). CM was supported by a grant from the Swiss Confederation (no 2015.0788).

**Competing interests:** The authors have declared that no competing interests exist.

because it is often associated with predator-prey relationships [22] and animal personality [16, 23–26].

Natal dispersal can be described as a three-stage process with two stationary stages (departure from the natal site and settlement) separated by an exploratory transient (or search) stage [27]. The first stage, departure, refers to the decision to leave the parental home-range; the transient stage includes dispersal movements and temporal settlement taking place between leaving the parental home range and the first breeding site. Individuals have to decide whether they quickly disperse to have access to high-quality territories [8, 28, 29] or if they wait until enough body reserves have been acquired and foraging skills developed [30, 31]. These are important decisions that affect survival and reproductive success [6].

Experience during early-life, which includes natal dispersal, shapes the adult phenotype in a complex way. It involves the individual's own genetic makeup and the environment experienced during development [32–34] which is sensitive to factors such as food supply [35, 36], weather conditions [37, 38], predation risk [39, 40] and sibling competition [41–43]. Indeed, energetic restrictions and other environmental perturbations during development may irreversibly affect the phenotype and have consequences for life [32, 44, 45] and may provoke a trade-off between survival and development. Glucocorticoids, such as corticosterone, play an essential role in resolving such trade-offs by modulating energy expenditure and behaviour to maintain body functions [reviewed in 46]. For instance, in response to unpredictable perturbations in the environment, corticosterone levels can rapidly increase to redirect resources towards essential functions, such as locomotion and foraging activity [47–51]. It has also been suggested that an increase in corticosterone prior to fledging initiates natal dispersal in birds [52].

In the present study, we explored the departure and the first part of the transient phase of dispersal in nestling barn owls (*Tyto alba*). Our goal was to study the impact of developmental stress on dispersal and survival. Although many aspects of the biology of this species have been well studied [53], the post-fledging period remains virtually unknown. However, two studies showed that the distance travelled in the first-year of life is heritable and related to melanin-based plumage coloration, with darker reddish owls moving longer distances than white individuals [13, 54]. Melanin-based plumage coloration is known to be associated with many physiological and behavioural traits [55] and individuals with larger black spots are less sensitive to stressful conditions compared to individuals with smaller spots [45, 48, 56, 57]. Here, we were interested in a different aspect of dispersal behaviour, namely in the early steps of natal dispersal taking place in the first few months after fledging. To this end, we modified post-natal rearing conditions through experimentally increasing corticosterone for a short period of time during the nestling stage [58]. This artificial increase in corticosterone levels temporally hampered growth but did not affect survival until fledging and body mass at fledging [45]. Here, we investigated whether these short-term stressful developmental conditions have long-lasting effects on natal dispersal behaviour and on post-fledging survival and whether these effects vary between individuals showing different melanin-based coloration.

## Methods

### Study species

In the study area, barn owls lay 2–11 eggs between February and August. The eggs hatch asynchronously on average every 2–3 days generating a pronounced within-brood age hierarchy. The altricial nestlings take their first flight at ca. 55 days [59]. Although plumage traits slightly change between the first and second year of age [60], variation in plumage coloration is already visible in nestlings. Although members of both sexes can show any phenotype, females

typically display on average a darker reddish-brown plumage (a phaeomelanin-based trait) marked with more and larger black spots (a eumelanin-based trait) than males. The expression of plumage traits is under strong genetic control and weakly sensitive to the rearing environment and body condition [61].

## Experimental design

The study was carried out in 2004 and 2005 in western Switzerland (46˚49′N /06˚56′E) in an area covering 190 km$^2$ where 150 nest boxes placed on barns were available for breeding barn owls. The nest boxes were regularly checked to monitor clutch sizes and hatching dates. Shortly after hatching, we measured wing length to determine nestling age [62]. Nestlings were ringed at the age of three weeks, i.e. before we implanted them with a corticosterone or placebo implant. The sex of all nestlings was determined using the CHD gene [63]. In both years, the nest boxes were controlled regularly to monitoring the growth of the juveniles. The controls took place around hatching, when the oldest nestling reached 3 weeks, at age 25–35*, age 28–32*, age 32–36*, age 41–43 and age 50* of the oldest nestling. During these controls the nestlings were measured and weighed and where indicated with an * blood samples to measure baseline and stress-induced corticosterone were taken [for details see 45, 58]. To investigate the effects of stressful physiological rearing conditions, mimicked through a moderate increase of corticosterone, we implanted the four oldest nestlings (in three broods only two nestlings hatched) in 2004, and the two oldest nestlings in 2005 (mean age ± SD at implantation, 29 ± 4 days; range, 21–35 days) with either a corticosterone or a placebo implant [see details in 58]. The corticosterone implants increased plasma corticosterone above baseline level for about 3 days within a physiological range [see details in 64]. The increase in plasma corticosterone due to the corticosterone implant (before implantation 17.4 ng/ml ± 16.7 (SD), two days after implantation 29.5 ng/ml ± 16.7, [for details on the methods see 58, 64]) was below an increase in corticosterone as a response to handling (67.9 ng/ml ± 16.1) and comparable with 32 h of food deprivation (20ng/ml ± 17 ng/ml, 45).

We obtained post-fledging movement data of 44 corticosterone-implanted juveniles (hereafter cort-juveniles; 22 females and 11 males in 2004 and 4 females and 7 males in 2005) and 45 placebo-implanted juveniles (hereafter placebo-juveniles; 21 females and 12 males in 2004 and 7 females and 5 males in 2005), and from 6 non-implanted nestlings (that were pooled with placebo-juveniles) from a total of 31 broods in 2004. In 2005, we also tagged the fathers from 12 of the 31 broods, which was useful to examine when the juveniles start to become independent from the father who is the main food provider. We used radio-transmitters with a weight of 8 g (< 3% of body mass at fledgling) and batteries with a lifespan of at least nine months. We attached the transmitters with a leg-loop harness (rubber band) on the back [65] when nestlings were on average 49 days old (cort-juveniles: 49.6 days (range 43–59), placebo-juveniles: 49.3 days (41–59)). At this age juveniles were not yet able to fly which we confirmed by checking whether the transmitter signals are still coming from inside the nest boxes the day following the transmitter attachment. None of the juveniles had left the nest at this age. The rubber band broke after roughly one year and the transmitters fell off. Following the day of transmitter attachment, we tracked each individual periodically at their diurnal roost sites using a three-element Yagi antenna and hand-held VHF receiver (R-1000, Communications Specialists Inc., CA, USA). Juveniles born in 2004 were relocated from 22 June to 4 December each second day until September, every fourth day in October and November, and every 7–10 days in December. In 2005, juveniles and adult males were relocated from 13 July to 14 December, every 2–4 days until September and once per month from October till December. The type of roost and their geographical location was successfully registered on 3117 occasions

for a total of 95 juveniles aged 50 to 227 days. For sixteen individuals (13 females and 3 males) we never located a roost outside their nest box until we lost contact at the age of 60 to 80 days. Until the end of the study period, 17 juveniles were found dead (6 cort-females and 6 cort-males, 4 placebo-females and 1 placebo-male) at the age 68 to 228 days. We registered 194 locations of roost sites from 16 radio-tracked males who were the fathers of 59 radio-tracked nestlings that were located during the same telemetry session.

## Plumage traits

AR recorded plumage traits and body measurements in all nestlings a few days before fledging. He placed a 60 × 40 mm frame on the breast, counted spots and measured the diameter of a large number of representative spots to the nearest 0.1 mm with a slide calliper, and calculated a mean spot diameter [for details of the method see 61, 66]. Note that within individuals the method of assessing plumage spottiness has already been shown to be reliable (repeatability is 0.93; [61]). Cort-nestlings displayed a similar number of spots as placebo-nestlings (73 ± 31 (sd) spots vs. 78 ± 27 spots; normal mixed-effect model with 'nest of origin' as random factor, p = 0.30). Because the corticosterone-experiment did not alter the expression of plumage traits [48, 67], we concluded that nestlings were randomly assigned to the two treatments with respect to plumage spottiness. AR also recorded the extent to which nestlings are reddish-brown with colour varying from white (score -8) to dark reddish (-1). This trait was recorded on the breast, belly, flank, and underside of one wing with the mean of these four body parts being used in the statistical analyses. Assessing this plumage trait is also reliable (repeatability is 0.95; [68]). The number of black spots and spot diameter are correlated (Pearson r = 0.62) and the association between number of black spots or spot diameter and the dependent variable in the models was quantitatively the same (same direction of correlation and similar effect size) and we therefore only used spot diameter in our final analyses. Further, in preliminary analyses, the degree of reddish coloration was not significantly associated with dispersal behaviour, and hence we do not consider it anymore in our final analyses.

All methods described in this study were approved by the cantonal committee for animal research (animal experiment permit no 1736 from the Veterinarian Office of Vaud).

## Statistical procedures

Since dispersal behaviour is affected by many individual traits, we included in the statistical models brood size, position in the within-brood age-hierarchy (first-born individuals had rank 1, second-born rank 2, and so on) and body size as measured by tarsus length (at fledging, when tarsus length has reached its final size) and spot diameter. We did not assess body condition, because barn owls lose body mass before taking their first flight implying that at that age, it is difficult to know whether a heavy or light body mass reflects poor condition or developmental stage. All measurements were taken blind to the experimental treatment.

## Nest departure and roost sites

To determine which factors were associated with the age when juveniles started to roost outside their rearing nest, we performed a binomial linear mixed model. We used the variable roost inside the nest box (1, with 1331 records) or outside the nest box (0, with 1650 records) as dependent variable and included nestling identity (95 juveniles) nested in rearing brood identity as random effects. We included year of birth, age, hatching date, sex, brood size, nestling rank, tarsus and wing length before fledging, spot diameter and implant (cort vs. placebo) as well as spot diameter of the biological parents as explanatory variables. It often happened that juveniles roosted outside their rearing nest box on one night and inside it on the following

night. For this reason, we calculated both the age when nestlings roosted outside the rearing nest box for the first time and inside it for the last time.

## Post-fledging movement

To estimate the age when juveniles leave for the first time their parental home-range (defined as an area of 1.5 km around the rearing site which corresponds to the home-range of breeding males, [69]), we performed a mixed–effect cox proportional hazards regression model using the same predictor variables as in the GLMM model, and brood identity as random factor.

We modelled the square-root transformed dispersal distance between the nest and the positions where an individual was recorded with respect to juvenile age, the experimental treatment and phenotypes of the juveniles with a generalized additive mixed model (GAMM) using the same predictor variables as above (year of birth, spot diameter, tarsus length and the corticosterone-implant in interaction with nestling age), and the same random effects.

We evaluated whether juveniles started to disperse while their father was still in the surroundings of their nest. We performed a normal linear mixed model with the distance between the diurnal roost of the juvenile and that of its father on the same day as the dependent variable as a function of the distance between the diurnal roost of the juvenile and its place of birth, with father and juvenile identity as random effects. We also investigated whether the bond between father and offspring decreased in strength when the offspring started to disperse and hence became independent from the parents. To this end, we ran two general additive mixed models (GAMM), one with the distance between the diurnal roost of the juvenile and its father as dependent variable as a function of the age of the juvenile and one with the distance between the roost of the father and its breeding place as dependent variable as a function of the age of the juvenile. In both models father and juvenile identity were included as random variables. We had data for 59 juveniles and 16 fathers collected in 2005.

Dispersal direction was calculated using rearing nest box as starting point and the point where each owl was located for the last time. We applied a Rayleigh's test to test whether the direction of juvenile dispersal is non-random [70] using all 79 juveniles that were located at least 3 km from the rearing nest box and then by considering subsets of individuals that were recorded the last time at a relatively old age, i.e. 100 days (47 juveniles), 110 days (35 individuals) and 120 days (31 individuals) of age.

## Post-fledging survival

To evaluate whether corticosterone treatment influenced juvenile survival, we performed a mixed-effect Cox proportional hazards regression model, using hatching date, position in the within-brood age hierarchy, tarsus length, and corticosterone-implant as independent variables and rearing brood identity as random effect.

Explanatory continuous variables were centred and scaled by their standard deviation before analysis. We verified that the assumptions of the models were met by examining residual diagnostic plots, which was always the case. All models were performed with the statistical software R version 3.3.2 [71] using the function glmmPQL of the MASS package [72] for GLMM model, mgcv package [73] for the GAMM model, adehabitatLT package [74] to estimate the angles, and circular package [75] for Rayleigh's tests, and coxme package for Cox model with random effects.

## Results

### Nest departure and roost sites

Nestlings roosted for the first time outside their rearing nest box during the daylight hours at a mean age of 75 days (range: 59–105 days, no significant difference between cort- (76 days ± 9 (sd)) and placebo-individuals (73 days ± 9)). The last time nestlings roosted in their rearing nest box was at a mean age of 79 days (range: 60–113 days, no significant difference between cort- (81 days ± 12) and placebo-individuals (76 days ± 11)). The very first diurnal roost site registered outside nest boxes was located inside barns for 25 individuals (36%), in trees at the forest edge for 18 individuals (25%), in isolated trees for 14 individuals (18%) and in groups of trees (but not in forests) for 13 individuals (16%). Trees remained an important roosting site until around age of 120–130 days, after this age the use of trees as roosting site steadily decreased while barns were again more often chosen (Fig 1). Adult males used almost entirely barns as sleeping places (99.4%, n = 194 sleeping places of 16 males) and only in one case a tree.

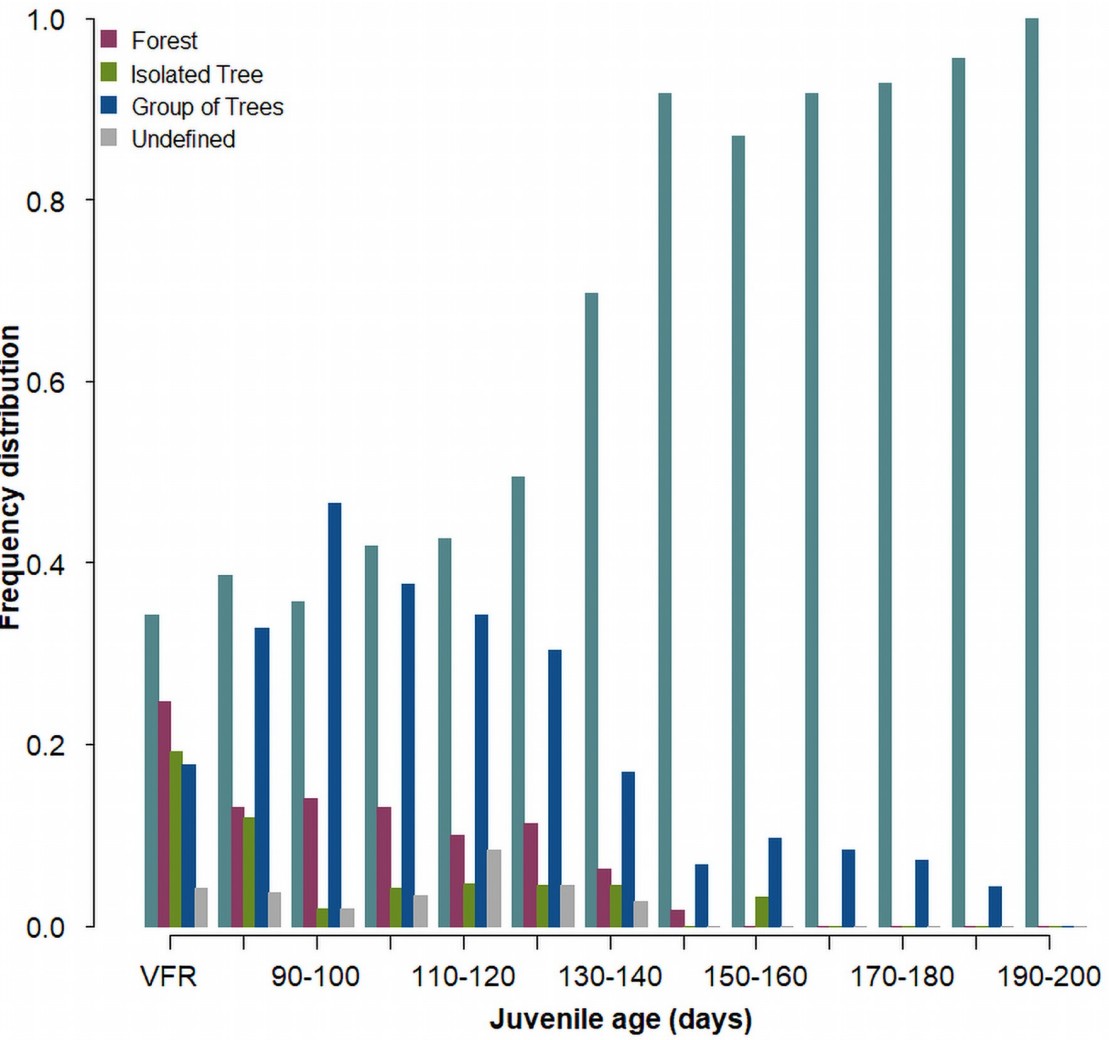

**Fig 1. Frequency distribution of diurnal roost sites for 95 juvenile barn owls found for the very first time (VFR) outside their nest box in 79 individuals and between age 80 and 190 days in 70 individuals totalling 1529 observations.**

**Table 1. Binomial linear mixed model describing the probability to roost in the rearing nest-box depending on year of birth (2004 or 2005), sex, age, implant (corticosterone and placebo), and diameter of black feather spots in juvenile barn owls.**

| | Estimate (se) | df | t value | p value |
|---|---|---|---|---|
| Intercept (male, corticosterone) | -2.4 (0.68) | 2884 | -3.49 | <0.001 |
| Year of birth (2005) | 0.59 (0.62) | 29 | 0.96 | 0.345 |
| Sex (female) | 0.68 (0.34) | 61 | 2.01 | 0.049 |
| Age | -4.70 (0.20) | 2884 | -23.44 | <0.001 |
| Age^2 | 0.75 (0.18) | 2884 | 4.17 | 0.001 |
| Implant (placebo) | -0.70 (0.24) | 61 | -2.86 | 0.006 |
| Spot diameter | -0.24 (0.18) | 61 | -1.37 | 0.177 |
| | **Variance** | | | |
| Nest of rearing | 1.48 | | | |
| Individual identity | 0.72 | | | |
| Residual | 1.31 | | | |

Individual identity and nest of rearing were introduced as random intercepts. The model is based on 2981 measurements of 95 individuals from 31 broods obtained in 2004 and 2005. The explicatory variables were centred and scaled by their standard deviation, and the temporal autocorrelation considered (Phi: 0.05).

Sixty-one per cent of the juveniles that were located outside their nest box during the daylight hours on one occasion were found back in their nest box on a subsequent day. The probability to roost during the day inside the rearing nest box decreased with age and was higher in juveniles implanted with a corticosterone pellet than with a placebo pellet (Table 1 and Fig 2).

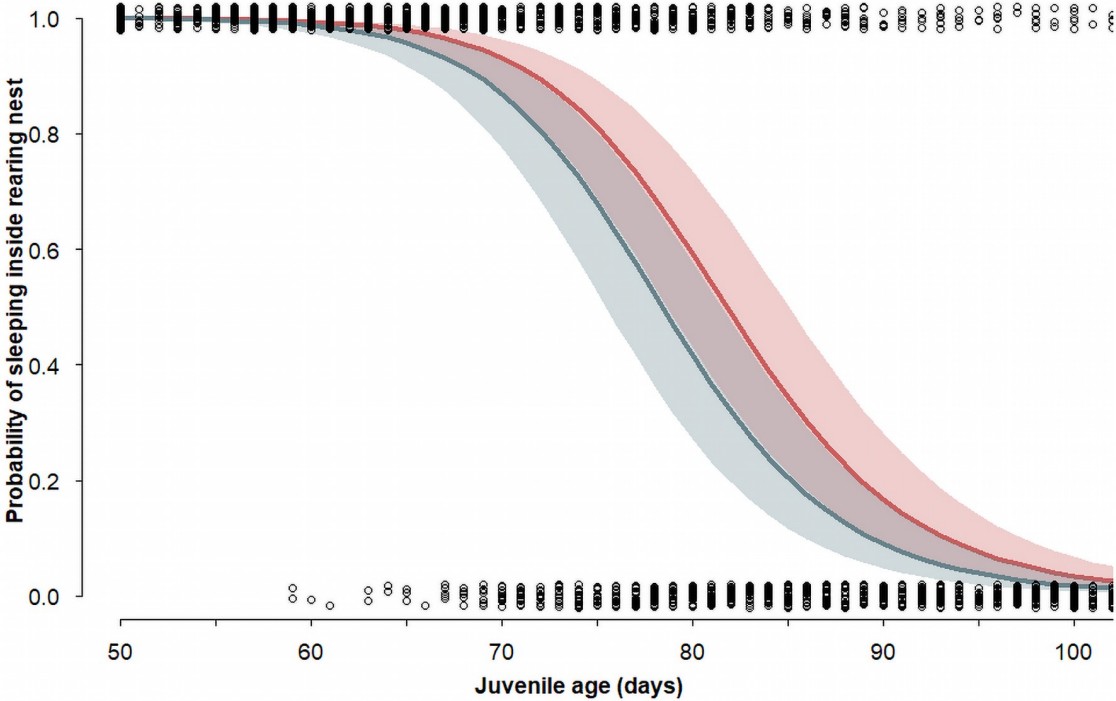

**Fig 2. Probability (with 95% CrI) of roosting inside the rearing nest-box in relation to age fitted with a binomial GLMM (Table 1) in juvenile barn owls.** The red line represents corticosterone-implanted individuals, the grey-green line placebo-implanted individuals, and circles the raw data.

Spot diameter was not related to the probability of roosting inside the rearing nest box (Table 1).

## Post-fledging movement

Juveniles started to leave the home range of the parents defined as an area of 1.5 km around the rearing nest box at an age of 97 ± 18 (mean ± sd) days (median 92 days, range 73–153; Fig 3). The age at which juveniles left the parental home range for the first time significantly differed between cort- and placebo-juveniles depending on the sex (β = 8.9 ± 0.6 SE, interaction treatment x sex: z = 2.08, p = 0.035). Placebo-females started to leave the parental home range at an earlier age than placebo-males (Fig 3), whereas cort-females left the parental home range at a later age than placebo-females but at a similar age as cort- and placebo males (Figs 3 and 4). Longer dispersal distances were reached by females compared to males, by individuals with larger spots compared to individuals with smaller spots (Fig 5a, 5b, 5e and 5f), by larger individuals as measured by tarsus length (Fig 5c, 5d, 5g and 5h) and by individuals experimentally treated with corticosterone rather than with a placebo implant (Table 2 and Fig 5). At an age of 5 month placebo-males reached a dispersal distance of 3.4 km (range: 2.3–4.8 km), placebo-females 9.6 km (7.9–11.5 km), which is a substantially shorter dispersal distance than the cort-group reached at this age (cort-males 5.2 km (3.7–6.8), cort-females 12.5 km (10.5–14.6 km); fitted estimates of the model presented in Table 2).

During the first six months post-fledging (until December) the distance between the daylight roosts of juveniles and of their father increased linearly in relation to the distance

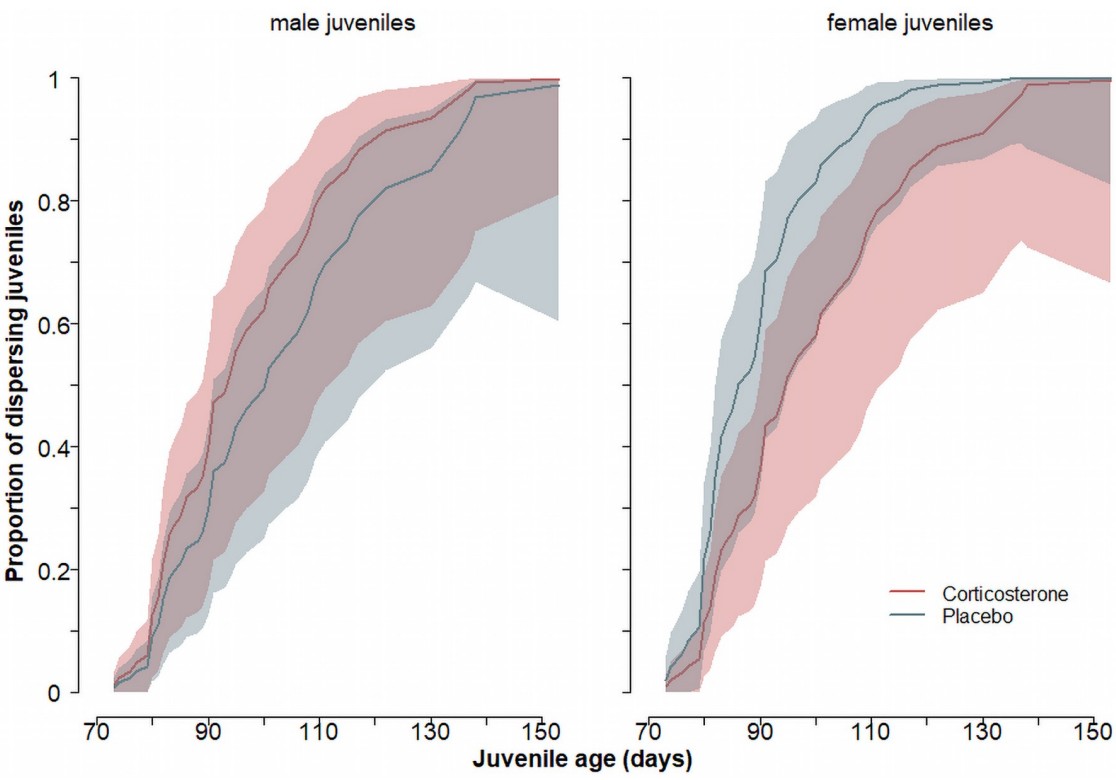

**Fig 3. Proportion of juveniles (with 95% CrI) leaving the parental home range in relation to juvenile age based on a cox proportional hazard model.** Left panel: male juveniles, right panel: female juveniles. Corticosterone- and placebo-implanted juveniles are in red and grey-green, respectively.

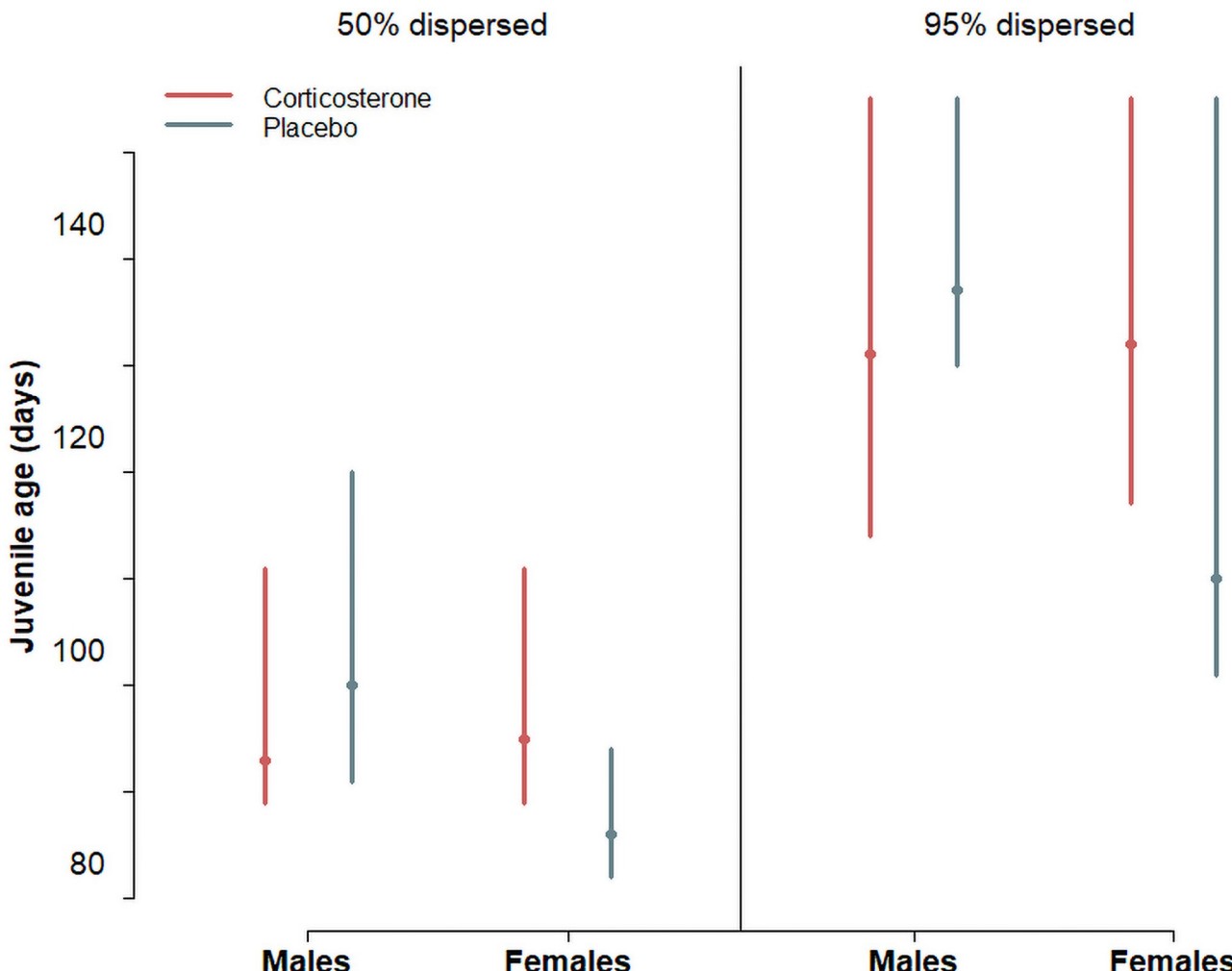

**Fig 4. Estimated age (with 95% CrI) when 50% (left panel) and 95% (right panel) of the juvenile barn owls left the parental home range based on a cox proportional hazard model.** Corticosterone- and placebo-implanted juveniles are in red and blue, respectively.

juveniles were located from their rearing nest box (Fig 6). Besides, the distance between father and its offspring increased with the age of the juvenile, while the distance between the father and the rearing nest box increased to much lower levels with the age of juvenile (Table 3 and Fig 7). At an age of four months the distance between the roost of the juvenile and of its father was 7.7 km (range 5.9–9.6 km) compared to the distance of 1.4 km (0.8–2.4 km) between the roost of the father and of its breeding place. This indicates that when a juvenile starts to leave the natal area, it does it without its father who stays close to its breeding place.

The distribution of dispersal directions did not differ significantly from random in any of the age classes when considering all individuals, those recorded after 100 days of age and in individuals having dispersed more than 3 km (Rayleigh's test, p-values > 0.5).

## Post-fledging survival

In total, 5 of 54 placebo-individuals and 12 of 44 cort-individuals died after nest-departure. Cort-juveniles had a lower probability to survive until the end of the year than placebo-

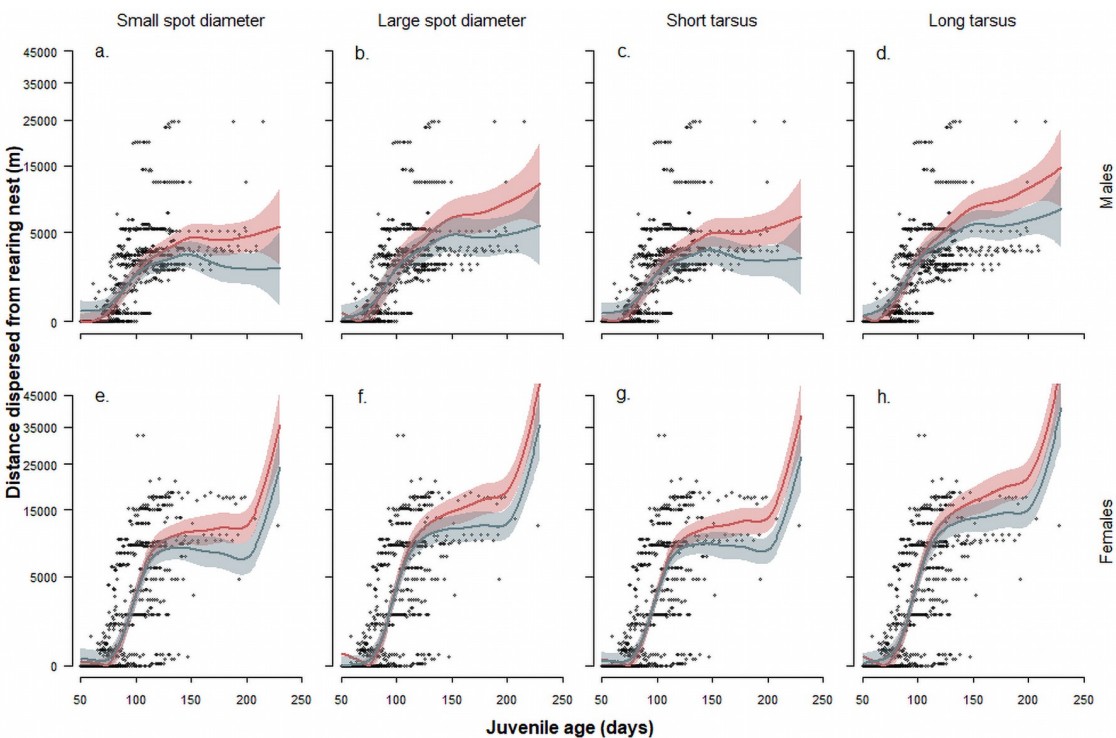

**Fig 5. Distance (with 95% CrI) between the nest of rearing and diurnal roost site in relation to age, treatment (placebo (blue) vs. corticosterone (red)), sex (males: panels a-d; females: panels e-h), spot diameter (small diameter: panels a, e; large diameter: panels b, f) and tarsus length (small: panels c, g; long: panels d, h) in juvenile barn owls.** Lines represent the group specific means with 95% CrI (shaded areas) based on the GAMM in Table 2. Black dots represent raw data.

**Table 2. Dispersal distance from the rearing nest in relation to year of birth (2004 or 2005), hatching date, age, sex (M = male, F = female), spot diameter, tarsus length and treatment (P = placebo, C = corticosterone) in juvenile barn owls.**

| fixed terms | Estimate (se) | t value | p value |
|---|---|---|---|
| Intercept (male, corticosterone) | 25.79 (3.8) | 6.7 | <0.001 |
| Year of birth (2005) | 5.67 (4.6) | 1.2 | 0.219 |
| Hatching date | 10.26 (3.7) | 2.7 | 0.006 |
| Sex (female) | 11.4 (4.1) | 2.8 | 0.006 |
| Implant (placebo) | 0.7 (3.7) | 0.2 | 0.858 |
| Spot diameter | 2.1 (4.0) | 0.5 | 0.607 |
| Tarsus length | 9.6 (3.7) | 2.6 | 0.009 |
| Tarsus length x Age | 20.0 (2.4) | 8.4 | <0.001 |
| Spot diameter x Age | 11.9 (2.2) | 5.3 | <0.001 |
| Implant (placebo) x Age | -14.5 (2.1) | -7 | <0.001 |
| **smooth terms** | **edf** | **F** | **p value** |
| s(age) x Sex (male) | 6.1 | 86.4 | <0.001 |
| s(age) x Sex (female) | 8.3 | 248.8 | <0.001 |

Individual identity, nest of rearing and year were three random factors in a generalized additive mixed model (GAMM). The model is based on 3117 observations of 95 individuals from 31 families obtained in 2004 and 2005. The response variable was square-root transformed, the explicatory variables were centred and scaled by their standard deviation, and the temporal autocorrelation of order 1 considered (Phi: 0.9). R2 adj: 0.50.

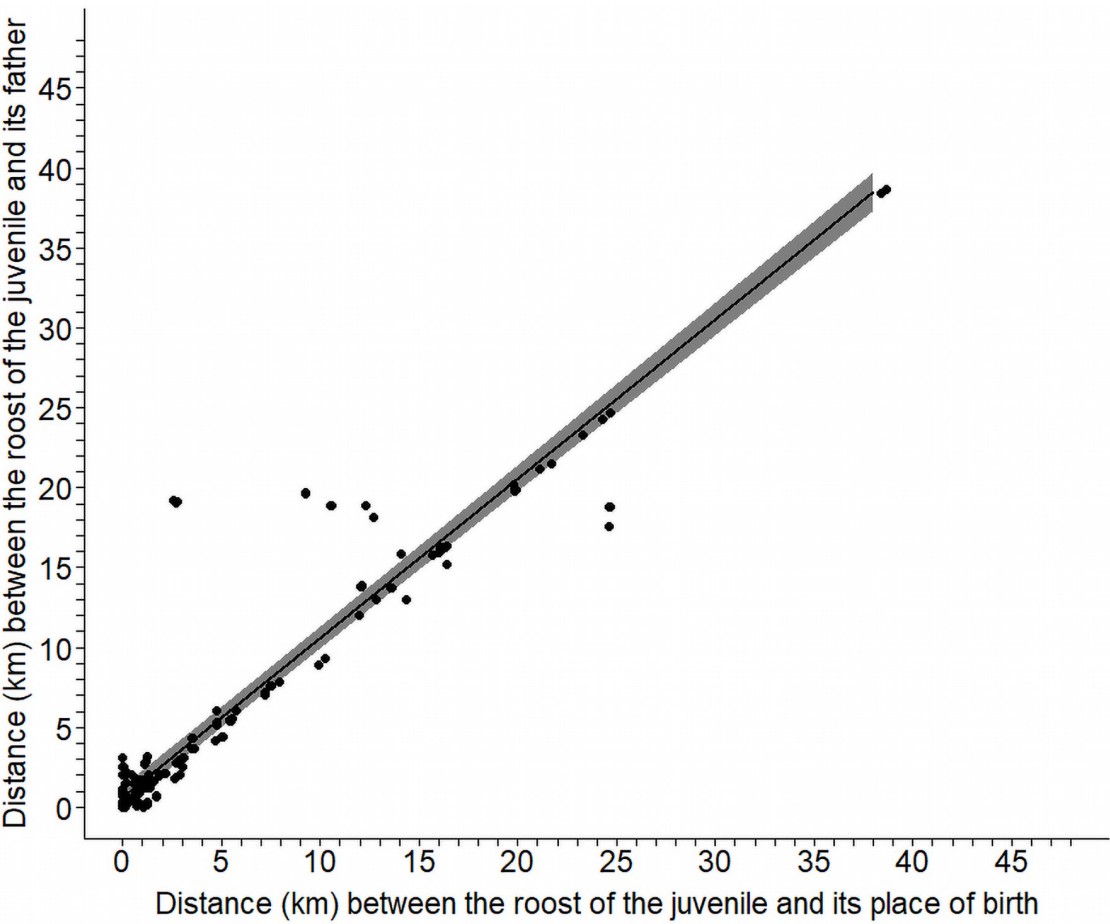

**Fig 6. Distance between the roost of the juvenile and its father in relation to the distance between the roost of the juvenile and its place of birth.** Shown are raw data (squares) and the fitted line with 95%CrI of a linear-mixed effect model with 'Distance between the roost of the juvenile and its father' as dependent variable, 'distance between the roost of the juvenile and its place of birth' as explanatory variable and juvenile identity nested in family as random effect. The analysis is based on 542 observations of 23 juveniles from 12 broods in 2005.

juveniles ($\beta$ = -1.4 $\pm$ 0.6 SE, z = -2.22, p = 0.026, Fig 8). Five juveniles died after collision with a car or power line (2 cort-, and 3 placebo individuals), one placebo-individual died in a slurry-tank, one cort-individual in a ventilation shaft. From ten we only found remains (feathers, carcasses) and could not determine the cause of death.

**Table 3. Results of a generalized additive mixed effect model (GAMM) with A) Distances between fathers and their offspring in relation to offspring age and B) Distance between fathers and their rearing nest boxes in relation to offspring age as dependent variables.**

| | A) Distance between the roost of the juvenile and ist father | | | B) Distance between the roost of the father and ist breeding place | | |
|---|---|---|---|---|---|---|
| Fixed terms | Estimate (se) | t value | p value | Estimate (se) | t value | p value |
| Intercept | 45.51 (4.26) | 10.06 | <0.001 | 25.89 (5.01) | 5.17 | <0.001 |
| Smooth terms | edf | F | p value | edf | F | p value |
| Age | 5.66 | 106.50 | <0.001 | 3.96 | 51.69 | <0.001 |

In the two models, offspring identity nested in father identity was a random factor. The models are based on 542 observations of 23 juvenile offspring of 12 fathers obtained in 2005.

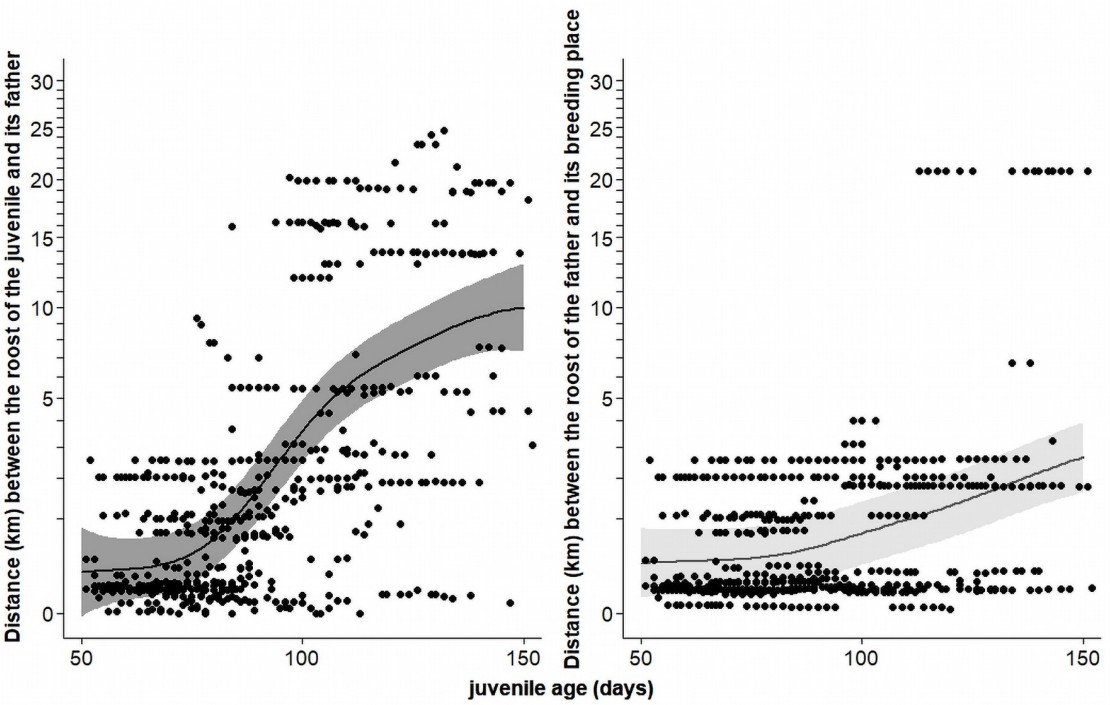

**Fig 7. Left panel: Distance between daylight roost of the juvenile and its father in relation to juvenile age. Right panel: distance between daylight roost of the father and its breeding place in relation to juvenile age.** The figure is based on the GAMMs presented in Table 3. Dots represent raw data and solid lines the population mean with 95%CrI. The analyses are based on 542 observations of 23 juveniles from 12 broods in 2005.

## Discussion

In the barn owl, the artificial administration of corticosterone elevated its blood-circulating levels during two to three days, and hence mimicked stressful rearing conditions only during this period of time [64]. This resulted in short-term effects on growth [45] and, in the present study, we report for the first time long-term effects. Cort-implanted nestlings delayed nest departure and implanted females left the parental home-range later than placebo-females, which was not the case in males. We also found that larger individuals, those with larger eumelanic spots on their plumage and cort-implanted individuals dispersed longer distances than small, smaller-spotted and placebo-individuals, respectively. Therefore, the timing and distance of the first step of dispersal from the natal territory, as well as the survival probability during the first months of life, depended on a short-term treatment of corticosterone. Dispersal behaviour is therefore related to phenotypic traits (melanin-based coloration and body size) but also to the level of this stress-hormone.

### Roost sites and post-fledging period

Juvenile barn owls roosted outside their nest box for the first time at around 60 days after hatching and they were regularly coming back to their nest box to roost until 110 days (Fig 2). Juveniles started to leave the parental home-range around 97 days of age when the distance between their daily roost and the daily roost of their fathers increased rapidly. Throughout the dispersal period the distance between the place where the juveniles roosted and their place of birth and the distance between the roost of the juvenile and its father's roost increased with

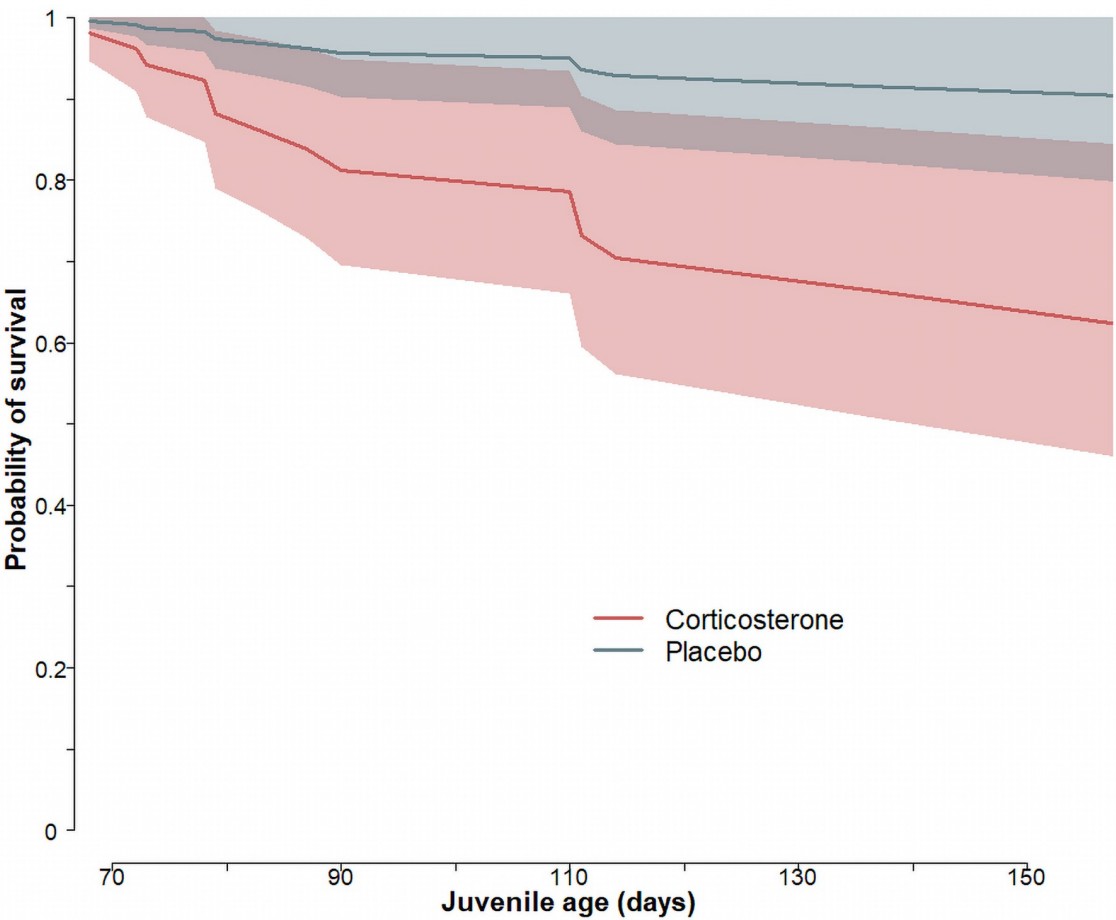

**Fig 8. Probability of survival (with 95% CrI) in relation to age in corticosterone-implanted (red) and placebo-implanted (grey-green) juvenile barn owls.**

age at the same pace, while the distance between father's roost and its breeding nest boxes increased only slightly. This indicates that the family break-up took place around the same time as the juveniles left the parental home-range, suggesting that after having left the nest box for the first time, juveniles are still dependent on their fathers for approximately one month, which is comparable to the length of the post fledging dependence period of other similar-sized and larger birds of prey [76–80]; but see [14] for a longer post-fledging period in Western screech owls (*Otus kennicottii*). Our observations also suggest that fathers feed their offspring only relatively close to their nest.

## Impact of stress on dispersal behaviour

Stressful physiological rearing conditions mimicked through a short period of increased plasma corticosterone levels resulted in a temporal decrease in body mass and temporally slowed down growth rate of wing length [45]. In the present study, we further showed that a temporal increase in the level of circulating corticosterone during the rearing period affected dispersal behaviour many weeks later: Individuals with a short-term corticosterone increase during the mid-rearing phase delayed the departure from the rearing nest and in females also

permanent emigration from the parental home range and then induced females to disperse longer distance. This demonstrates that the conditions experienced during the rearing phase not only impact juvenile development but have also longer lasting effects on dispersal behaviour later in life. The fact that cort-individuals left the parental site at a later time than placebo-individuals is similar to observations performed in the little owl (*Athene noctua*) where nestlings raised in poor-food habitats delayed their first foray and permanent emigration from the natal site [80]. The delayed departure from the nest box and rearing home range of cort-juvenile barn owls cannot be explained by poor body condition per se, since the experimental treatment of corticosterone, which led to a temporal body mass decrease, was fully caught up at fledging [45]. However, in some bird species an increase in corticosterone prior to fledging seems to trigger the onset of natal dispersal [81, 82]. In the present study, the corticosterone response to handling stress was still decreased due to the experimental treatment shortly before fledging [64], i.e. 14 days after the experimentally increased levels of blood circulating corticosterone returned to baseline. This indicates that the physiological condition of the juveniles was still affected by the experimental treatment, and could in turn delay nest departure. A similar study in wild tree swallows (*Tachycineta bicolor*) experimentally treated with corticosterone found no difference between treated and placebo individuals in the age at which juveniles departed from the nest, even though corticosterone-treated nestlings had a lower body mass [83]. There may be other factors that determine the timing of nest departure than body or physiological conditions in this species. A potential explanation is that in a species like the barn owl showing pronounced within-brood age hierarchy and a large number of siblings, individuals have a longer time window to restore adverse body condition. As long as the youngest nestlings are still fed by their parents, the older siblings have a chance to obtain food in the nest and can therefore delay nest departure. In species with less pronounced age hierarchy, individuals in poor condition might be forced to leave the nest sooner with the rest of the family to get food provisioning outside the nest. This idea is supported by a study in the barn swallow (*Hirundo rustica*) showing that the duration of the post-fledging period is more crucial for survival than the conditions experienced during the rearing period [84, 85].

Stressful rearing conditions delayed permanent emigration from the parental home range in female barn owls but not in males. Given that in birds females generally move longer distances than males, they may leave the nest at the earliest possible time while males do not lose anything by remaining a couple of days longer in the nest. This is also what we observed in the placebo-group: females permanently emigrated from the parental home-range earlier than males. Further, females dispersed more abruptly than males (Fig 3), as has already been observed in the Tengmalm's owl (*Aegolius funereus*) [17]. This does not preclude that the age at emancipation is sensitive to environmental conditions. Indeed, female juveniles treated with corticosterone delayed departure and ended up leaving the parental home range at a similar age as males. Cort-individuals dispersed further than placebo-individuals perhaps because of the need to evade stressful areas. For instance, after a breeding season, unsuccessful redshank (*Tringa tetanus*) females dispersed farther than successful females [86]. The barn owl could show similar patterns, as longer dispersal distances are attained after harsh winters [87] and when population size increases suggesting that dispersal may be triggered by intraspecific competition to find a territory to settle down [88].

## Sex-specific variation in dispersal behaviour

The distance between the daily roost of the juvenile and its place of birth increased gradually with their age until emancipation. This finding is similar to what was found in studies on other raptors (e.g. bald eagle (*Haliaeetus leucocephalus*): [89]; western screech-owls: [14]; northern

goshawk (*Accipiter gentilis*): [90]; eagle owl (*Bubo bubo*): [91, 92]; and Tengmalm's owl: [11]), where dispersal distance increases linearly until the end of the transient period of dispersal. As Ellsworth & Belthoff [14] pointed out, the distance measured in those cases does not represent the final dispersal distance. When we compare the dispersal distance reached by the juveniles until the end of the year in the present study to long term capture-recapture data of the same population [13], we see that dispersal distances of females (placebo-females 9.8 km, cort-females 12.8) were comparable to or even larger than natal dispersal distances found with capture-recapture data (11.7 km; [13]) while males still reached shorter dispersal distances (placebo-males 3.4 km, cort-males 5.2 km) than those found with capture-recapture data (8.7 km; [13]). In the present study, we followed radio-tagged individuals within and outside the study area in which we fixed nest boxes implying that we had more opportunities to locate owls far from their birth sites, particularly females who disperse longer distances than males. Accordingly, 13 of the 16 individuals that we could not relocate after they left the nest boxes were females, perhaps because they moved beyond the area that we prospected. This is likely because it is not rare that juveniles disperse over very long distances sometimes up to 1'000 km in Europe. In any case, our observations suggest that there is a difference in the timing of dispersal between the sexes with females reaching earlier their final natal dispersal area than males. That females already reached the area of their first breeding attempt at the end of the year is in line with studies performed in Germany [87] and in Denmark [93].

## Dispersal behaviour in relation to phenotype

We found that larger individuals dispersed further than smaller ones. Although this hypothesis is widely accepted at the species level [94–96], at the interspecific level this pattern remains unclear. Body size has been related to social status, with bigger individuals dominating smaller ones [14]. If dispersal is costly, dominant juveniles are expected to disperse shorter distances, which was not the case in the barn owl. Alternately, if there are advantages to long-distance dispersal, then the dominant individuals, which are in better condition, should disperse further than subordinate individuals [14]. In our case, we do not have any information about a potential link between dominance status and body size, and hence the link between body size and dispersal remains unclear. We also found that dispersal distance was related to the size of black feather spots. Large-spotted individuals dispersed further than smaller-spotted individuals. Previous studies in the barn owl showed that individuals with large spots breed at a younger age and have a higher survival [97], are less sensitive to stressful conditions (loose less weight when food deprived [57], have a lower corticosterone response to an acute stress situation [56] and are more resistant to oxidative stress [98]) than individuals with smaller spots. All these results suggest that individuals with large black spots might be better able to cope with the unpredictable situations they encounter during dispersal. However, two previous studies which looked at natal dispersal with ring recovery data could not find any relation between the final dispersal distance and the size of the black feather [13, 54]. This disparity between studies could be due to difference in timing rather than in the final dispersal distance in relation to black feather spots. Possibly individuals with larger black spots arrive earlier at the first breeding site and therefore also start to breed at a younger age, while final dispersal distance might not differ between individuals with varying size of black feather spots. This hypothesis should be further studied.

## Post-fledging survival

Barn owl juvenile's post-fledging survival was affected by a moderate increase of corticosterone during nestling development. Elevated corticosterone during the rearing period has the

potential to compromises cognitive abilities that are relevant for foraging success, nest defence, and predator avoidance [99] as well as immune functions [100, 101]. These impairments in the development could therefore compromise survival. Besides, experimentally increased corticosterone levels during development impact telomere length [83, 102, 103], and some studies have linked reduced survival with telomere shortening [102, 104–106]. However, the relationship between glucocorticoids and survival is not clear: artificially elevated glucocorticoids had no effect on tree swallow survival even in individuals showing telomere shortening [83], however decreased survival in black-legged kittiwakes (*Rissa tridactyla*) [107], while in Swainson's thrush (*Catharus ustulatus*) high baseline corticosterone has been associated with increased survival [108]. In the barn owl, a strong stress-response is positively associated with survival [109], while an increase in corticosterone levels in nestlings dampened their stress-response later just before fledging [64] probably indicating that an alteration of the HPA-axis in nestlings affected survival in the first year.

## Acknowledgments

We thank all the field assistants who helped us collect the data and follow the owls along their dispersal paths.

## Author Contributions

**Conceptualization:** Bettina Almasi, Lukas Jenni, Alexandre Roulin.

**Data curation:** Bettina Almasi.

**Formal analysis:** Bettina Almasi, Carolina Massa.

**Funding acquisition:** Carolina Massa, Lukas Jenni, Alexandre Roulin.

**Investigation:** Bettina Almasi.

**Methodology:** Bettina Almasi, Carolina Massa, Lukas Jenni.

**Project administration:** Bettina Almasi.

**Supervision:** Lukas Jenni, Alexandre Roulin.

**Validation:** Bettina Almasi.

**Visualization:** Bettina Almasi, Carolina Massa.

**Writing – original draft:** Bettina Almasi, Carolina Massa, Alexandre Roulin.

**Writing – review & editing:** Bettina Almasi, Carolina Massa, Lukas Jenni, Alexandre Roulin.

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
