## [Decision Letter · Decision Letter 0]

17 May 2021

PONE-D-21-07930

Exogenous corticosterone and melanin-based coloration explain variation in juvenile dispersal behaviour in the barn owl (Tyto alba)

PLOS ONE

Dear Dr. Almasi,

Thank you for submitting your manuscript to PLOS ONE. After careful consideration, we feel that it has merit but does not fully meet PLOS ONE’s publication criteria as it currently stands. Therefore, we invite you to submit a revised version of the manuscript that addresses the points raised during the review process.

An expert individual in the field has reviewed the manuscript and found it to be of interest. However, some points require further clarification, including cause of death in some cases.

We look forward to receiving your revised manuscript.

Kind regards,

Cheryl S. Rosenfeld, DVM, PhD

Academic Editor

PLOS ONE

Journal Requirements:

"The Swiss National Science Foundation supported financially the study (no 3100A0-

104134 to LJ and no PPOOAO-102913 to AR). CM was supported by a grant from the Swiss

Confederation (no 2015.0788)."

 "YES

no 3100A0-104134 to LJ

no PPOOAO-102913 to AR

no 2015.0788 to CM

http://www.snf.ch"

Reviewers' comments:

Reviewer's Responses to Questions

**Comments to the Author**

1. Is the manuscript technically sound, and do the data support the conclusions?

Reviewer #1: Yes

2. Has the statistical analysis been performed appropriately and rigorously? 

Reviewer #1: Yes

3. Have the authors made all data underlying the findings in their manuscript fully available?

Reviewer #1: No

4. Is the manuscript presented in an intelligible fashion and written in standard English?

Reviewer #1: Yes

5. Review Comments to the Author

Reviewer #1: I thank the authors for sharing an engaging, well-written manuscript detailing the factors that relate to juvenile dispersal in barn owls. In particular, the finding that a short-term physiological elevation of corticosterone delayed juvenile dispersal while ultimately leading to longer final dispersal distances in females and reducing survival is very interesting and makes an important contribution to this field. I have just a few questions/comments to help improve the paper, mostly about methods and analysis.

Re. timing of implants: I think the text indicates this on lines 107-109, but the implants were put in at an age of 3 weeks, correct?

At what age were the radio transmitters attached? Did the measurement of plumage traits and tarsus length, etc. a few days before fledging carry any risk of force fledging the nestlings? In addition, elsewhere another study is cited that mentions that these same nestlings underwent a stress series later during the nestling period – would it be possible to provide a timeline or other clear description in one place of the number of times that individuals were handled and/or sampled during the nestling period? Where the ages of handling/sampling the same in 2004 and 2005?

For the individuals that were found dead, was a cause of death apparent or discernable? This seems especially important given the difference between CORT and placebo-treated birds.

And finally, I noted that there were some differences between the 2 years of sampling (smaller sample sizes in 2005, different numbers of birds treated as nestlings in each nest each year, as well as different frequencies of radio tracking post-fledging). However, year is not included in any of the statistical models as far as I can tell – given these differences in experimental design, I’m a bit concerned that the results may have differed between years, because of year effects or because of design differences. I’d suggest incorporating some kind of analysis that considers year/design effects.

6. PLOS authors have the option to publish the peer review history of their article (what does this mean?). If published, this will include your full peer review and any attached files.

Reviewer #1: No

---

## [Author Response · Author response to Decision Letter 0]

29 Jun 2021

Dear Cheryl S. Rosenfeld

Thank you for giving us the opportunity to resubmit our manuscript. Below you find our answers to the questions raised by the reviewer.

Sincerely, on behalf of the co-authors, 

Bettina Almasi

Reviewer #1: I thank the authors for sharing an engaging, well-written manuscript detailing the factors that relate to juvenile dispersal in barn owls. In particular, the finding that a short-term physiological elevation of corticosterone delayed juvenile dispersal while ultimately leading to longer final dispersal distances in females and reducing survival is very interesting and makes an important contribution to this field. I have just a few questions/comments to help improve the paper, mostly about methods and analysis.

Re. timing of implants: I think the text indicates this on lines 107-109, but the implants were put in at an age of 3 weeks, correct?

Our response: the nestlings received the implants at a mean age of 29 days; details are listed on line 116ff. 

At what age were the radio transmitters attached? 

Our response: Indeed, this information is missing. We added the following on line 132ff.: We attached the transmitters with a leg-loop harness (rubber band) on the back (Naef-Daenzer 2007) when nestlings were on average 49 days old (cort-juveniles: 49.6 days (range 43 – 59), placebo-juveniles: 49.3 days (41 – 59)). 

Did the measurement of plumage traits and tarsus length, etc. a few days before fledging carry any risk of force fledging the nestlings? 

Our response: There is no risk of force fledging if the nestlings are younger than 55 days of age (personal experience). However, at this visit to the nest boxes we will close the nest entrance with a net for the rare cases when nevertheless a nestling tries to jump out of the nest boxes or when the oldest nestlings are older than 55 days. Before putting the nestlings back to the nest box we close the nest entrance with a sponge attached to a long rope. A few minutes after the nestlings have been set back to the nest box we pull the rope and observe the entrance for a while. It never happens that any juvenile jumps/flies out. We added the following explanation on line 133ff: At this age juveniles were not yet able to fly which we confirmed by checking whether the transmitter signals are still coming from inside the nest boxes the day following the transmitter attachment. None of the juveniles had left the nest at this age. 

In addition, elsewhere another study is cited that mentions that these same nestlings underwent a stress series later during the nestling period – would it be possible to provide a timeline or other clear description in one place of the number of times that individuals were handled and/or sampled during the nestling period? Where the ages of handling/sampling the same in 2004 and 2005?

Our response: We added the following information on line 108: In both years, the nest boxes were controlled regularly to monitoring the growth of the juveniles. The controls took place around hatching, when the oldest nestling reached 3 weeks, at age 25-30*, age 28-32*, age 32-36*, age 41-43 and age 50* of the oldest nestling. During these controls the nestlings were measured and weighed and where indicated with an * blood samples to measure baseline and stress-induced corticosterone were taken (for details see Almasi et al. 2012, Almasi et al. 2009).

For the individuals that were found dead, was a cause of death apparent or discernable? This seems especially important given the difference between CORT and placebo-treated birds.

Our response: We added the following information in the result section on line 280ff.: Five juveniles died after collision with a car or power line (2 cort-, and 3 placebo individuals), one placebo-individual died in a slurry-tank, one cort-individual in a ventilation shaft. From then we only found remains (feathers, carcasses) and could not determine the cause of death.

And finally, I noted that there were some differences between the 2 years of sampling (smaller sample sizes in 2005, different numbers of birds treated as nestlings in each nest each year, as well as different frequencies of radio tracking post-fledging). However, year is not included in any of the statistical models as far as I can tell – given these differences in experimental design, I’m a bit concerned that the results may have differed between years, because of year effects or because of design differences. I’d suggest incorporating some kind of analysis that considers year/design effects.

Our response: Thank you of pointing this out. We now added the variable ‘Birth year’ in all analysis which included data of both years. Please note that we never detected any significant effect of ‘birth year’ and the results did not change whether we included birth year or not.

---

## [Decision Letter · Decision Letter 1]

29 Jul 2021

Exogenous corticosterone and melanin-based coloration explain variation in juvenile dispersal behaviour in the barn owl (Tyto alba)

PONE-D-21-07930R1

Dear Dr. Almasi,

We’re pleased to inform you that your manuscript has been judged scientifically suitable for publication and will be formally accepted for publication once it meets all outstanding technical requirements.

Kind regards,

Cheryl S. Rosenfeld, DVM, PhD

Section Editor

PLOS ONE

Additional Editor Comments (optional):

Reviewers' comments:

Reviewer's Responses to Questions

**Comments to the Author**

1. If the authors have adequately addressed your comments raised in a previous round of review and you feel that this manuscript is now acceptable for publication, you may indicate that here to bypass the “Comments to the Author” section, enter your conflict of interest statement in the “Confidential to Editor” section, and submit your "Accept" recommendation.

Reviewer #1: All comments have been addressed

2. Is the manuscript technically sound, and do the data support the conclusions?

Reviewer #1: Yes

3. Has the statistical analysis been performed appropriately and rigorously? 

Reviewer #1: Yes

4. Have the authors made all data underlying the findings in their manuscript fully available?

Reviewer #1: Yes

5. Is the manuscript presented in an intelligible fashion and written in standard English?

Reviewer #1: Yes

6. Review Comments to the Author

Reviewer #1: I thank the authors for their careful attention to my comments, and am happy to recommend this article for publication in Plos One.

One quick note: for the additional information on nest checks (line 108), I'd suggest that the authors state that "nest boxes were checked regularly to monitor the growth..." -- using the term "checked" instead of "controlled" for clarity.

7. PLOS authors have the option to publish the peer review history of their article (what does this mean?). If published, this will include your full peer review and any attached files.

Reviewer #1: No

---

## [Editor Report · Acceptance letter]

20 Aug 2021

PONE-D-21-07930R1 

Exogenous corticosterone and melanin-based coloration explain variation in juvenile dispersal behaviour in the barn owl (*Tyto alba*) 

Dear Dr. Almasi:

I'm pleased to inform you that your manuscript has been deemed suitable for publication in PLOS ONE. Congratulations! Your manuscript is now with our production department. 

Kind regards, 

on behalf of

Dr. Cheryl S. Rosenfeld 

Section Editor

PLOS ONE